# Excision of Intramedullary Osteoid Osteomas in the Posterior Tibial Area via Medulloscopy: A Case Report

**DOI:** 10.3390/medicina57020163

**Published:** 2021-02-12

**Authors:** Jong Hoon Park, Hae Woon Jung, Woo Young Jang

**Affiliations:** 1Department of Orthopedic Surgery, Anam Hospital, Korea University College of Medicine, Seoul 02841, Korea; pjh1964@hanmail.net; 2Department of Pediatrics, Kyung Hee University Medical Center, Seoul 02841, Korea; woonieya@gmail.com

**Keywords:** osteoid osteoma, medulloscopy, patient-specific instrument

## Abstract

Osteoid osteomas are benign bone-forming lesions that usually present in adolescence. In patients with severe pain and those not responding to medication, surgical treatment should be considered. Medulloscopy is a standard arthroscopic technique for visualizing the intramedullary canal of the tibia. Herein, we report two patients with intramedullary osteoid osteomas in the posterior area of the tibia, which were successfully treated using medulloscopy. Hence, medulloscopy is an effective minimally invasive method in patients with intramedullary osteoid osteomas in the posterior tibial area.

## 1. Introduction

Osteoid osteomas, initially described by Jaffe in 1935, are common benign bone tumors occurring in patients under 25 years of age. These tumors are associated with nocturnal pain that is markedly reduced by the administration of nonsteroidal anti-inflammatory drugs. Osteoid osteomas elicit a profound osteoblastic response in the surrounding medullary and cortical bone and show a characteristic picture of sclerosis around a lucent nidus [1]. On plain radiographs, an osteoid osteoma appears as a small central core of lower density within a dense cortical bone in the diaphysis, surrounded by reactive sclerosis [2]. These clinical and radiographic findings lead to the diagnosis of osteoid osteoma. Surgical therapy should be considered when severe pain is not relieved with conservative management such as medication. For surgical therapy, conventionally, en bloc resection or curettage has been used. Recently, alternative methods, such as radiofrequency ablation and computed tomography (CT)-guided percutaneous excision, have also been used [3].

Osteoid osteomas can be classified according to their location as cortical, cancellous, or subperiosteal [1]. Medulloscopy has been used to visualize lesions in the intramedullary canal of the femur or tibia [4,5]. Several authors have advocated for the use of medulloscopy for osteomyelitis lesions. However, there have been no reports of intramedullary osteoid osteoma excision using medulloscopy [4,5]. Herein, we report two patients with intramedullary osteoid osteomas in the posterior area of the tibia, which were successfully treated with medulloscopy. Treatment procedures for the patients were carried out in accordance with the Helsinki Declaration and were approved on 4 November 2020 by the Ethics Committee of Korea University Anam Hospital with code number 2020AN0486.

## 2. Case Presentations

### 2.1. Case 1

A 13-year-old girl presented with right lower leg pain, which she had been experiencing for the past 2 months. Plain radiographs showed a well-circumscribed lesion with a partially calcified nidus (Figure 1A,B). CT showed a well-defined oval nidus with low attenuation (Figure 1C–E). Therefore, she was diagnosed with osteoid osteoma.

The operation was performed under general anesthesia with the patient lying in the supine position with a tourniquet. The nidus was identified using fluoroscopy. Then, two guide pins were inserted into the center of the lesion under fluoroscopic guidance. A 7.3-mm cannulated drill bit was used to drill into the cortex, and a hole was created for entry of the arthroscope (Figure 1F,G). One portal was used as the viewing portal for the arthroscope, and the other was used as the working portal for the instrument (Figure 1H). Arthroscopic findings showed a mineralized nidus lesion (Figure 1I). The nidus was excised for biopsy, and removal was performed using a curette and arthroscopic burr (Figure 1J). Postoperative CT revealed the complete resection of the nidus (Figure 1L–N), and the diagnosis of osteoid osteoma was confirmed by histological examination.

Partial weight bearing with crutches was allowed for the first 3 weeks. The patient’s symptoms resolved completely after the operation, and she was pain-free at the 1-month follow-up. She returned to sporting activities at 2 months and showed good functional scores (Lysholm score: 81, Western Ontario and McMaster Universities Osteoarthritis Index (WOMAC) score: 0).

### 2.2. Case 2

A 19-year-old man visited our clinic with right lower leg pain, which he had been experiencing for the past 1 month. Physical examination revealed a limping gait. Plain radiographs and CT images revealed a radiolucent center with surrounding reactive sclerosis on the posteromedial aspect of the tibial shaft (Figure 2).

Exact targeting of the medulloscope is needed for intramedullary lesions because of the small range of motion of the instrument due to cortical canal hardness. Therefore, we created a guide device using three-dimensional (3D) printing to target localized lesions on the posterior side of the tibia (Figure 3A,B). In the operating room, we easily accessed the nidus using this targeting guide and removed the nidus using medulloscopy (Figure 3C–F) with minimal incisions (Figure 3G). Postoperative CT showed complete resection of the nidus (Figure 3I,K).

After surgery, the patient’s leg was immobilized using a splint for the first 3 weeks to prevent possible pathologic fracture from the medulloscopy canal. Partial weight bearing with a single crutch was allowed for the next 3 weeks. At 3 months after surgery, CT showed good union of the cortex of the canal area, and the patient returned to sporting activities with good functional scores (Lysholm score: 94, WOMAC score: 0).

## 3. Discussion

Osteoid osteomas account for approximately 11% of benign bone tumors and frequently occur in children and young adults [1]. Metaphyses or diaphyses of long bones are the most frequent sites. Although non-operative treatment may be considered for lesions in challenging anatomical locations, the continued presence of osteoid osteomas can lead to growth disturbances such as limb length discrepancies and scoliosis in children, because the growth plate can be affected by severe inflammation caused by the osteoid osteoma [6]. In this study, a minimally invasive method using medulloscopy for adolescent patients with intramedullary osteoid osteomas showed successful outcomes.

There are two different approaches for surgical treatment: conventional surgical treatment, and percutaneous destruction of the nidus using several methods such as percutaneous sclerosis using radiofrequency ablation or thermocoagulation [3]. Conventional surgical therapy has several limitations. Open resection can result in a large bone defect, which may further require bone grafting and internal fixation. Therefore, to prevent postoperative fracture, immobilization with a splint after the operation is usually applied for long periods with restrictions on postoperative activities and weight bearing. Furthermore, intraoperative localization of the tumor may be challenging, leading to partial removal and potential recurrence. For these reasons, minimally invasive procedures have become the treatment of choice owing to their high rates of treatment success similar to that of open surgery. However, oval-shaped osteoid osteomas with a high eccentric index showed symptomatic recurrence following radiofrequency ablation. Furthermore, CT-guided procedures cannot excise the nidus for biopsy. For differentiation from inflammatory lesions such as osteomyelitis, a firm histological diagnosis is occasionally necessary. In this study, the preoperative radiographs in Case 2 suggested other diagnoses, such as osteoblastoma, Brodie abscess, and fracture deformity due to stress fracture, but it was histologically confirmed after surgery to be osteoid osteoma.

Medulloscopy was initially suggested as an arthroscopic instrument applied to the intramedullary canal [4]. Medulloscopy allows direct visualization, irrigation, and debridement of intramedullary lesions. Arthroscopic excision in critical anatomical sites, such as in the intra-articular locations of osteoid osteomas, has been reported with successful treatment outcomes. In this study, arthroscopic instruments were used for intramedullary osteoid osteomas in the posterior area of the tibia. For exact targeting of the nidus, a custom-designed device was created using 3D printing in Case 2. According to previous reports, average residual errors of less than 1° and 1 mm were reported for simulated osteotomies when using patient-specific guides [7]. A patient-specific guide device created by 3D printing also aided in exact targeting, which allowed easy access to the lesion in this study. Three months after medulloscopy, both patients returned to sporting activities.

## 4. Conclusions

In conclusion, medulloscopy is an effective minimally invasive method in patients with intramedullary osteoid osteoma at the posterior tibial area.

## Figures and Tables

**Figure 1 medicina-57-00163-f001:**
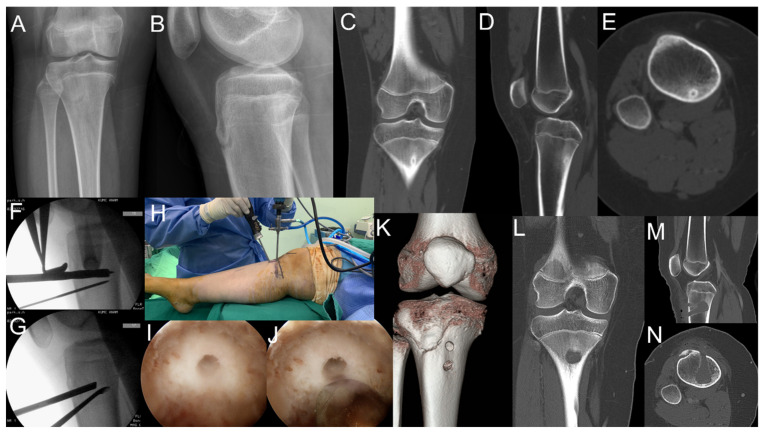
A 13-year-old girl presented with an intramedullary osteoid osteoma in the posterior area of the proximal tibia (**A**–**E**). She underwent excision of the nidus using medulloscopy (**F**–**I**). Complete removal of the nidus was confirmed through postoperative computer tomography (**J**–**N**).

**Figure 2 medicina-57-00163-f002:**
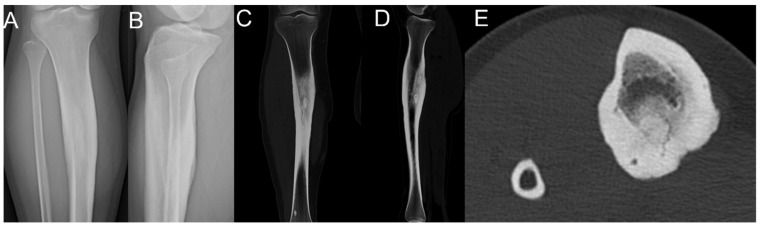
A 19-year-old man presented with an intramedullary osteoid osteoma at the diaphyseal tibial posterior area on plain radiographs (**A**,**B**). CT revealed a radiolucent center with surrounding reactive sclerosis on the posteromedial aspect of the tibial shaft (**C**–**E**).

**Figure 3 medicina-57-00163-f003:**
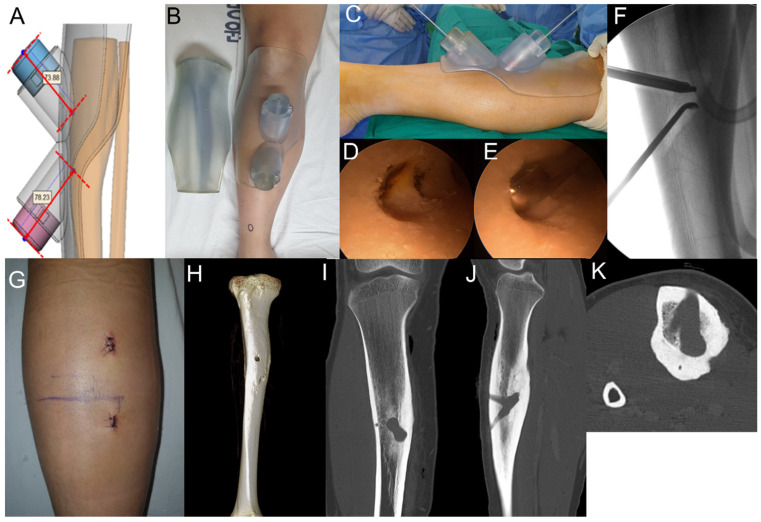
A patient-specific guide was created using 3D printing (**A**,**B**), and medulloscopy was performed (**C**–**F**) with minimal incision (**G**). Complete removal of the nidus was confirmed through postoperative computer tomography (**H**–**K**).

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
