# Peer review of "Excision of Intramedullary Osteoid Osteomas in the Posterior Tibial Area via Medulloscopy: A Case Report"

_medicina, 2021, doi:10.3390/medicina57020163_

Round 1

Reviewer 1 Report

Dear Woo Young Jang,

I suggest you increase the paper, it is a case report, but it is useful to explain to readers the topic with a short review of the current literature, with the use of recent references.

Best regards

Giuseppe Minervini

Author Response

Response to the Reviewers

Dear Reviewers,

We deeply appreciate your comments and questions, and believe that they have led to significant improvements in the manuscript.

The reviewers’ comments or questions were reiterated in italics, and we addressed them point-by-point. All changes in the revised manuscript are shown using the track change function in Microsoft Word.

Reviewer 1.

I suggest you increase the paper, it is a case report, but it is useful to explain to readers the topic with a short review of the current literature, with the use of recent references.

-> Thank you for your comment. According to your suggestion, we added the following sentence in the Introduction section:

Osteoid osteomas, initially described by Jaffe in 1935, are common benign bone tumors occurring in patients under 25 years of age. These tumors are associated with nocturnal pain that is markedly reduced by the administration of nonsteroidal anti-inflammatory drugs (NSAIDs). Osteoid osteomas elicit a profound osteoblastic response in the surrounding medullary and cortical bone and shows a characteristic picture of sclerosis around a lucent nidus [1]. These clinical and radiographic findings lead to the diagnosis of osteoid osteoma.  

Reviewer 2.

This manuscript by Jang and colleagues reports that the two cases of intramedullary osteoid osteomas in the posterior area of the tibia are successfully treated by using medulloscopy. The symptoms, diagnosis, treatment, and follow-up check are clearly described. It is a good addition to the treatment of osteoid osteoma.

-> We deeply appreciate for your comments.

Reviewer 2 Report

This manuscript by Jang and colleagues reports that the two cases of intramedullary osteoid osteomas in the posterior area of the tibia are successfully treated by using medulloscopy. The symptoms, diagnosis, treatment, and follow-up check are clearly described.  It is a good addition to the treatment of osteoid osteoma.

Author Response

(The authors gave the same response as above.)
